# Regulation of Airway Smooth Muscle Cell Proliferation by Diacylglycerol Kinase: Relevance to Airway Remodeling in Asthma

**DOI:** 10.3390/ijms231911868

**Published:** 2022-10-06

**Authors:** Miguel Angel Hernandez-Lara, Santosh K. Yadav, Sushrut D. Shah, Mariko Okumura, Yuichi Yokoyama, Raymond B. Penn, Taku Kambayashi, Deepak A. Deshpande

**Affiliations:** 1Center for Translational Medicine, Division of Pulmonary, Allergy and Critical Care Medicine, Jane & Leonard Korman Respiratory Institute, Sidney Kimmel Medical College, Thomas Jefferson University, Philadelphia, PA 19107, USA; 2Department of Pathology and Laboratory Medicine, Perelman School of Medicine, University of Pennsylvania, Philadelphia, PA 19104, USA

**Keywords:** airway smooth muscle, airway remodeling, proliferation, diacylglycerol kinase, asthma

## Abstract

Airway remodeling in asthma involves the hyperproliferation of airway smooth muscle (ASM) cells. However, the molecular signals that regulate ASM growth are not completely understood. Gq-coupled G protein-coupled receptor and receptor tyrosine kinase signaling regulate ASM cell proliferation via activation of phospholipase C, generation of inositol triphosphate (IP_3_) and diacylglycerol (DAG). Diacylglycerol kinase (DGK) converts DAG into phosphatidic acid (PA) and terminates DAG signaling while promoting PA-mediated signaling and function. Herein, we hypothesized that PA is a pro-mitogenic second messenger in ASM, and DGK inhibition reduces the conversion of DAG into PA resulting in inhibition of ASM cell proliferation. We assessed the effect of pharmacological inhibition of DGK on pro-mitogenic signaling and proliferation in primary human ASM cells. Pretreatment with DGK inhibitor I (DGKI) significantly inhibited platelet-derived growth factor-stimulated ASM cell proliferation. Anti-mitogenic effect of DGKI was associated with decreased mTOR signaling and expression of cyclin D1. Exogenous PA promoted pro-mitogenic signaling and rescued DGKI-induced attenuation of ASM cell proliferation. Finally, house dust mite (HDM) challenge in wild type mice promoted airway remodeling features, which were attenuated in DGKζ^-/-^ mice. We propose that DGK serves as a potential drug target for mitigating airway remodeling in asthma.

## 1. Introduction

Asthma is a chronic allergic airway disease. Repeated exposure to allergens and chronic inflammation in asthmatics leads to structural changes in lung tissues collectively known as airway remodeling (AR) [1,2,3]. Thickening of the airway wall due to excessive accumulation of airway smooth muscle (ASM) is a major component of AR, and clinical studies have established a direct correlation between the thickness of ASM mass and the severity of asthma [4,5,6]. Although significant strides have been made in developing asthma therapeutics based on immunological features, pharmacological approaches to mitigate features of AR including thickening of ASM are lacking [3]. Delineating mechanisms that promote ASM cell proliferation is the first critical step in establishing a therapeutic target to alleviate bronchoconstriction due to the thickening of ASM.

Previous studies from our lab have demonstrated that agonists of Gq-coupled G protein-coupled receptors (GPCRs) promote ASM proliferation [7,8,9]. In addition, growth factors promote ASM cell proliferation via the activation of receptor tyrosine kinases (RTKs) and mitogen-activated protein kinases (MAPKs) [7,10,11,12,13,14,15,16]. Activation of Gq-coupled GPCRs and RTKs on ASM cells leads to phospholipase C (β or γ)-mediated hydrolysis of phosphoinositide bisphosphate (PIP_2_) into inositol 1,4,5 trisphosphate (IP_3_) and diacylglycerol (DAG) [17]. IP_3_ elicits signaling events and physiological responses in ASM cells by elevating intracellular calcium concentration. DAG binds to protein kinase C (PKC), leading to activation of ERK1/2 MAPK, which in turn promotes ASM cell proliferation [7,18]. Signaling and functional effects of DAG are terminated by the conversion of DAG into phosphatidic acid (PA), a reaction catalyzed by diacylglycerol kinase (DGK) enzyme [19]. A recent study from our lab has demonstrated that pharmacological inhibition, or genetic ablation of ζ (or α) isoform of DGK, ameliorates allergen-induced airway hyperresponsiveness in an allergic asthma mouse model [20]. Furthermore, our studies have demonstrated that inhibition of DGK isoforms inhibits Gq-coupled GPCR agonist-mediated ASM contraction by regulating Gq signaling in a negative feedback manner, or by promoting paracrine release of pro-relaxant prostaglandin E2 [21,22]. Interestingly, PA is a phospholipid second messenger produced by DGK known to promote signaling of its own to regulate a variety of cellular functions, including proliferation and migration of cells [23,24,25,26].

A PA-induced mitogenic effect was demonstrated in HEK293 cells, and signaling studies further demonstrated that PA-induced mitogenesis involves activation of mTORC1 signaling and its downstream substrates [27,28]. In other studies, overexpression of DGKζ in HEK-293 cells induced PA-mediated activation of mTORC1 signaling, establishing a role for the ζ isoform of DGK in regulating cell growth and proliferation [29]. Another isoform, DGKη was implicated in C2C12 cell proliferation as it is abundantly expressed in this cell-type. In these studies, the authors knocked down DGKη expression via siRNA and reported decreased C2C12 proliferation and mTOR protein expression, suggesting a role for DGK-mediated activation of mTOR signaling (via PA) in C2C12 cell proliferation [30]. These studies implicate DGK isoforms and PA in regulating cell proliferation. However, the role of DGK and PA in the regulation of ASM cell proliferation is not known.

In this study, we hypothesized that PA is a pro-mitogenic second messenger in ASM cells and DGK inhibition results in the attenuation of growth factor-induced cell proliferation via depletion of PA levels in ASM cells. Treatment of human ASM cells with DGK inhibitor I (DGKI) resulted in time- and concentration-dependent inhibition of platelet-derived growth factor (PDGF)-induced ASM cell proliferation. Treatment of ASM cells with exogenous PA promoted ASM cell proliferation and restored ASM cell proliferation attenuated by DGKI. Western blot and luciferase assay analyses revealed that PA-mediated pro-mitogenic signaling involves activation of the mTORC1 signaling pathway, followed by activation of the transcription factor serum response element (SRE), and expression of cell cycle genes. Collectively, our data establish PA as a pro-mitogenic second messenger in ASM cells, and DGKI attenuates growth factor-mediated ASM cell proliferation via depletion of PA levels. House dust mite challenge in wild type mice resulted in multiple airway structural changes consistent with airway remodeling, and DGKζ knockout mice displayed significantly reduced features of airway remodeling. Coupled with our previous study, our findings demonstrate that DGK inhibition provides relief from multiple features of asthma pathogenesis, including airway inflammation, hyperresponsiveness, and remodeling.

## 2. Results

### 2.1. DGK Inhibition Reduces Human ASM Cell Proliferation

To determine the effect of DGK inhibition on ASM cell proliferation, we assessed total DNA content as a readout of cell proliferation using the CyQuant assay. Human ASM cells were pretreated with increasing concentrations of DGK inhibitor I ((DGKI) 10 μM–30 μM), followed by stimulation with PDGF (10 ng/mL), and DNA content was assessed at 24, 48, and 72 h. Treatment of human ASM cells with DGKI resulted in a time- and concentration-dependent inhibition of PDGF-induced ASM cell proliferation. At 72 h, treatment of human ASM cells with 10-, 20-, and 30-μM DGKI reduced PDGF-induced ASM cell proliferation by ~20-, 50-, and 70%, respectively (Figure 1a–c). We further tested cell viability in the presence of DGKI (10 μM–30 μM) for 24 h using MTT Cell Viability assay. Triton-X (1% vol.) was used as a positive control and data were normalized to this condition. Our findings demonstrate that in the presence of DGKI there is no significant difference in cell viability compared to basal, vehicle-treated conditions (Figure 1d). These findings demonstrate the anti-mitogenic effect of DGK inhibition in human ASM cells.

### 2.2. Exogenous PA Promotes Human ASM Cell Proliferation and Reverses the Anti-Mitogenic Effect of DGK Inhibition

DGK promotes the conversion of DAG into PA and DGK inhibition decreases the production of PA in ASM cells [19,22]. In HEK293 cells, PA induces pro-mitogenic signaling mediated by mammalian target of rapamycin (mTOR) signaling pathway [27]. Thus, we hypothesized that PA is a pro-mitogenic second messenger in ASM cells and depletion of PA by DGK inhibition results in attenuation of ASM cell proliferation. Human ASM cells were treated with increasing concentrations of PA (0.1 μM–10 μM) and total DNA content was determined after 72 h. Treatment of human ASM cells with PA significantly enhanced DNA content, demonstrating the pro-mitogenic effect of PA on human ASM cells (Figure 2a).

We further tested the ability of exogenous PA to overcome the anti-mitogenic effect of DGK inhibition. Human ASM cells were pretreated with DGKI (15 μM), followed by stimulation with increasing concentrations of PA (0.1 μM–10 μM) for 72 h. DGK inhibition resulted in reduced cell proliferation and 5 and 10 μM PA treatment reversed the growth inhibitory effect of DGKI (Figure 2b). Additionally, we tested the ability of PA to overcome the anti-mitogenic effect of DGKI in the presence of PDGF. Human ASM cells were treated with PA (0.1 μM–10 μM) for 10 min and then stimulated with PDGF (10 ng/mL) ± vehicle or DGKI (15 μM) for 72 h. Our data demonstrate a reversal of DGKI-induced inhibition of ASM cell proliferation by increasing concentration of PA (Figure 2c), and PDGF alone was used as a positive control. Together, these data support our hypothesis that PA is a pro-mitogenic second messenger in human ASM cells, and inhibition of DGK attenuates ASM proliferation by depleting PA levels in human ASM cells.

### 2.3. DGK Inhibition Does Not Influence PDGF-Induced MAP Kinase and PI3 Kinase Signaling

Previous studies have established that growth factors, such as PDGF, promote human ASM cell proliferation via activation of ERK MAPK and PI3K signal transduction pathways [7,12,14,15]. In addition, the absence of DGKζ in T lymphocytes results in enhanced ERK activation and T lymphocyte hyperactivation [31,32]. To assess the effect of DGK inhibition on PDGF-induced activation of ERK and PI3K signaling in human ASM cells, cells were treated with DGKI (30 μM) for 15 min followed by stimulation with PDGF (10 ng/mL) for 15 min, and whole cell lysates were subjected to immunoblot analysis. PDGF stimulation resulted in enhanced p-Akt (Figure 3a,b) and p-ERK1/2 (Figure 3c,d); however, DGK inhibition did not further significantly modulate p-Akt or p-ERK1/2 by PDGF (Figure 3a–d). These data demonstrate that DGK inhibition does not modulate activity of these canonical pro-mitogenic pathways in ASM cells.

### 2.4. DGK Inhibition and Exogenous PA Modulate mTOR Signaling

Next, we aimed at establishing the molecular target of DGK- and PA-mediated modulation of ASM cell proliferation. We hypothesized that PA promotes ASM cell growth via the mTORC1 signaling pathway and determined the effect of DGK inhibition, as well as exogenous PA stimulation, on mTORC1 activity in human ASM cells [28,33]. Human ASM cells were stimulated with increasing concentration of PA (1 μM–20 μM) for 20 min, and phosphorylation of mTOR (Ser2448) and its downstream substrate S6K was assessed by western blotting (Figure 4a). There was a significant increase in p-mTOR (Ser2448) and p-S6K (Figure 4b,c) by treatment with PA in a concentration-dependent manner. PDGF-induced activation of mTOR signaling was used as a positive control. Next, human ASM cells were pretreated with DGKI (30 μM) for 15 min followed by stimulation with vehicle, PDGF (10 ng/mL), or PA (10 μM) for 10 min. Cells were lysed, and phosphorylation of mTOR (Ser2448) and S6K was assessed by western blotting (Figure 4d). DGK inhibition reduced PDGF-mediated p-mTOR (Ser2448) and p-S6K, which was significantly restored by the addition of PA (10 μM) (Figure 4e,f).

To further establish the involvement of mTORC1 pathway in DGK and PA-mediated regulation of ASM cells proliferation, we pretreated human ASM cells with rapamycin (10 nM), a potent mTORC1 inhibitor, followed by stimulation with PA (0.5 µM–10 µM). Rapamycin attenuated PA-mediated p-mTOR (S2448) (Figure 5a,b) and blunted p-S6K (Figure 5a,c). We performed co-immunoprecipitation studies to further establish the mechanism by which PA regulates activation of mTORC1. Prior studies suggest that PA and rapamycin compete for the FRB domain on mTOR, leading to activation or inhibition of mTORC1 complex formation, respectively [27,34]. Human ASM cells were pretreated with vehicle or rapamycin (10 nM) for 10 min followed by stimulation with vehicle, PDGF (10 ng/mL), or PA (5 or 10 μM) for 20 min. PA induced mTOR (Ser2448) phosphorylation and formation of mTORC1 by increased recruitment of raptor to the complex (Figure 5d). Rapamycin (10 nM) reduced formation of this complex, and phosphorylation of mTOR. PDGF (10 ng/mL) was used as a positive control. Together, these studies demonstrate that PA induces activation of the mTORC1 pathway by promoting formation of the complex, and inhibiting DGK activity attenuates this signaling mechanism by depleting levels of PA.

### 2.5. PA Activates Transcription Factor Serum Response Element

mTOR activation regulates cellular functions by activation of transcription factors which in turn regulate gene expression profiles. We investigated the effect of exogenous PA on activation of transcription factors SRE, NFAT, Smad, and AP-1 in human ASM cells using a luciferase (Luc) reporter [35]. There was a significant increase in SRE-Luc activity upon PA (5 and 10 μM) stimulation of human ASM cells (Figure 6) with no effect on the activity of other transcription factors (data not shown). To validate that the activation of SRE is mediated by PA-induced mTORC1 activation, we pretreated cells with or without rapamycin (10 nM) for 10 min and then stimulated with vehicle, PDGF (10 ng/mL), or increasing concentrations of PA (0.5 μM–10 μM) for 12 h. Rapamycin significantly reduced PA-mediated induction of SRE-Luc activity (Figure 6) compared to control. These findings suggest that PA enhances SRE transcription factor activity via mTORC1.

### 2.6. DGK Inhibition and PA Modulate the Expression of Pro-Mitogenic Genes

Cell proliferation is regulated by cell cycle regulating proteins, thus, we sought to determine the expression of cell cycle genes affected by DGK inhibition and PA treatment. Human ASM cells were stimulated with PDGF (10 ng/mL) or PA (10 µM) for 72 h, followed by RNA extraction and analysis of cell cycle genes using a qRT-PCR array. A total of 25 and 45 genes were upregulated more than two-fold by PDGF and PA treatment, respectively, compared to vehicle-treated cells. Among these, 20 genes were upregulated by both PA and PDGF (Figure 7a). Interestingly, 25 unique genes were upregulated by PA, suggesting unique mechanisms activated by PA in human ASM cells to promote cell proliferation. A list of genes differentially enriched by PDGF or PA treatment in human ASM cells is included in Figure 7b. Human ASM cells pretreated with DGKI (30 µM), followed by stimulation with PDGF (10 ng/mL) downregulated 16 genes, out of which expression of 11 genes was rescued by treatment with PA (10 µM). A list of all genes upregulated by PDGF that were attenuated by DGK inhibition, and then rescued by PA along with their role in cell cycle progression, is presented in Table 1.

### 2.7. PA Increases Cyclin D1 Levels Attenuated by DGK Inhibition

We further validated the involvement of cell cycle genes in DGK-mediated mTOR signaling regulation of ASM cell proliferation by determining the expression of cyclin D1, a well-known cell cycle regulator in ASM cells. Human ASM cells were incubated with rapamycin (10 nM) for 10 min and then stimulated with PA (5 μM or 10 μM) for 24 h, lysed, and total protein was used to assess the expression of cyclin D1 by western blotting. There was a significant increase in cyclin D1 expression in human ASM cells upon PA (10 μM) stimulation (Figure 8a,b) compared to basal vehicle-treated condition. Rapamycin pretreatment inhibited the expression of cyclin D1. Further, human ASM cells were pretreated with DGKI (30 µM), followed by stimulation with PDGF (10 ng/mL) ± PA (10 μM) for 24 h. Cells were lysed and cyclin D1 expression was assessed (Figure 8c,d). Cyclin D1 expression was significantly attenuated by DGKI and restored by exogenous PA (Figure 8c,d). These data suggest that DGK inhibition results in reduction of human ASM cell proliferation by attenuation of transcriptional regulation of cell cycle genes due to depletion of PA, a pro-mitogenic second messenger mediating mTORC1 signaling.

### 2.8. DGKζ KO Mice Exhibit Reduced Expression of Airway Remodeling Markers upon HDM Challenge

To assess the in vivo effect of DGK inhibition on features of airway remodeling, we utilized our previously characterized global DGKζ^−/−^ mouse model [20]. Mice were challenged using HDM allergen for 3 weeks and lung tissues were collected at the end of the challenge. Whole lung tissue sections were stained for cellular markers of airway remodeling. Our data demonstrate increased airway inflammation (Figure 9a), mucus cell metaplasia (Figure 9b), proliferation of lung cells (Figure 9c), and thickening of ASM layer (Figure 9d) in HDM-challenged wild type mice. These markers were significantly attenuated in the airways of HDM-challenged DGKζ^−/−^ mice. These in vivo findings implicate DGKζ in the development of allergen-induced airway remodeling.

## 3. Discussion

Excessive proliferation of ASM cells is a cardinal feature of asthma, yet current asthma management does not address this feature. One potential unexplored target is DGK. The rationale for exploring the therapeutic potential of DGK stems from the central role of DGK isoforms in the regulation of lipid second messengers, DAG and PA, which are pivotal in pro-mitogenic and pro-contractile signaling in ASM cells. In fact, a recent study from our lab demonstrated that DGK knockout mice are resistant to the development of allergen-induced airway inflammation and hyperresponsiveness. Additional cell-based studies further established the regulation of contractile signaling by DGK in human ASM cells. In this study, we sought to investigate the effect of DGK inhibition on human ASM cell proliferation and establish the pro-mitogenic signaling mechanism(s) regulated by phospholipids in ASM cells. Treatment of human ASM cells with DGKI resulted in a time- and concentration-dependent inhibition of PDGF-induced ASM cell proliferation. Stimulation of human ASM cells with exogenous PA promoted cell proliferation via mTORC1 signaling pathway activation, and reversed DGKI-induced anti-mitogenic effect. Collectively, our data demonstrate that PA is a pro-mitogenic second messenger in human ASM cells and DGK inhibition attenuates growth factor-mediated proliferation by depleting PA levels. Further, our mechanistic studies show that PA activates mTORC1 signaling with downstream activation of the transcription factor, SRE, and promotion of cell cycle via increased expression of cyclin D1 in ASM cells (Figure 10).

DAG produced upon activation of Gq-coupled GPCR and RTK signaling activates PKC and MAPK signaling pathways [17,36]. DAG is converted to PA by the action of a lipid kinase, DGK, and the functional role of PA in ASM cells has not been thoroughly investigated. PA is a phospholipid second messenger that is known to regulate a variety of cellular functions, including cell proliferation, migration, and differentiation [23,24,25,26]. Stimulation of ASM cells with histamine increased the accumulation of different species of DAG and PA, and DGK inhibition decreased agonist-induced PA levels in ASM cells [22]. Agonist-induced accumulation of PA in different types of smooth muscle has been demonstrated previously. Carbachol stimulation increased PA levels in longitudinal intestinal and airway smooth muscle [37,38], and angiotensin, or noradrenaline, stimulation promoted the accumulation of PA in vascular smooth muscle [39,40]. An increase in the intracellular concentration of PA also occurs upon stimulation of hepatic stellate cells with platelet-derived growth factor [24]. The formation of PA has been linked to a variety of physiological responses such as actin polymerization, cell migration, and mitogenesis, [23,24,25,41] functions relevant to smooth muscle cell physiology. Findings from the present study demonstrate that PA is a pro-mitogenic second messenger in ASM cells. In agreement with our findings, PA is known to promote proliferation of osteoblastic cells in response to mitogenic stimuli [42].

Having established the pro-mitogenic effect of PA, we sought to explore the mechanistic basis for the pro-mitogenic effect of PA in ASM cells. In this context, our immunoblot studies demonstrate that DGK inhibition does not modulate ERK MAPK and PI3K, two predominant pro-mitogenic signaling mechanisms in ASM cells (Figure 3). This led us to explore the other signal transduction targets in ASM cells that are potentially regulated by DGK and PA. Our study demonstrates that PA-induced ASM proliferation involves activation of mTOR signaling, followed by activation of transcription factor SRE, resulting in the upregulation of a variety of cell cycle regulatory genes. In agreement with our findings, PA-induced activation of mTOR signaling has been demonstrated in other cell types [27,34,43]. In mouse extensor digitorum longus muscle, PA treatment increased mTOR activity, which was abolished by pretreatment with rapamycin [43]. PA is known to induce proliferation via activation of mTORC1 and not mTORC2 [28]. mTORC1 activation is regulated by growth factors, amino acids and other nutrients, and its physiological cellular functions are mediated via phosphorylation of its substrates, S6K1 and 4E-BP1, involved in protein synthesis and growth [27,33,44]. mTORC2 is regulated by nutrients and amino acids. mTORC2 plays a role in cell growth, metabolism, survival, and cytoskeletal reorganization via activation of its substrates Akt (Ser 473) and cPKC [45], whereas mTORC1 is formed with the association of rapamycin sensitive protein, raptor, and mTORC2, associated with the rapamycin insensitive protein, rictor. We demonstrated that PA promotes increased phosphorylation of mTOR and its substrate S6K, which are both inhibited by pretreatment with rapamycin. Co-immunoprecipitation studies give further insight on the increased formation of mTORC1, and support our hypothesis that PA stimulates cell proliferation by activating the mTOR signaling pathway. Our results also demonstrate the regulation of cell cycle gene expression in ASM cells by PA, including Cyclin D1 protein. Consistent with our findings, PA has been shown to be produced by DGK activation in T cell, inhibiting DGK results in the arrest of the cell cycle in the late G1 phase, thereby attenuating T cell growth [46].

DAG produced upon Gq-coupled GPCR activation directly activates PKC family members and Ras guanyl nucleotide-releasing protein, which leads to downstream signaling pathways involving NF-κB, ERK1/2, and Akt [17]. PKC phosphorylates multiple targets in ASM cells and regulates contraction, gene expression, and proliferation. PKC activation promotes calcium- and calcium sensitization-mediated contraction of ASM cells [47]. Different isoforms of PKC are known to regulate expression of cell cycle genes including Cyclin D1 in ASM cells, thereby promoting cell proliferation [18]. It is expected that DGK inhibition enhances DAG levels in ASM cells, thereby promoting ASM cell proliferation. Contrary to this, our data demonstrate that inhibition of DGK results in the attenuation of ASM cell proliferation. The relationship between PKC and DGK involving DAG is very complex. Increased DAG leads to activation of PKC which subsequently induces activation of DGKζ isoform, promoting the conversion of DAG to PA [19]. This suggests that PKC and DGK activation tightly regulate cellular DAG levels. Furthermore, PA is also involved in the regulation of PLC activity in a feedback manner [48]. Thus, DGK inhibition could work through reduced PA production and DAG-mediated negative feedback inhibition of PLC activity to reduce the proliferation of ASM cells.

There are ten known isoforms of DGK and DGK-mediated activation of mTOR by PA generation, which are isoform-specific. mTOR activation by DGK was seen in HEK cells overexpressing DGKζ, but not DGKα [29]. Similarly, mechanical stimulation-induced PA-mediated activation of mTOR signaling has been shown to involve DGKζ isoform in skeletal muscle. DGKζ activity sufficiently induces skeletal muscle fiber hypertrophy through mTOR, and in DGKζ knockout mice there was a reduced mechanically-induced increase in PA-mediated activation of mTOR signaling. These results indicate that DGKζ isoform plays an important role in mechanically-induced increases in skeletal muscle mass [49]. In line with these studies, our findings from the knockout mice suggest that DGKζ plays a critical role in the regulation of airway cell proliferation and allergen-induced airway remodeling.

In summary, our findings identify PA as a novel regulator of ASM proliferation, and inhibition of DGK represents a promising therapeutic approach to mitigate features of ASM remodeling in asthma. Combined with our murine model of asthma studies [20], findings from the current study establish DGK as a novel therapeutic target with a potential inhibitory effect on multiple features of asthma, namely airway inflammation, remodeling, and hyperresponsiveness.

## 4. Materials and Methods

### 4.1. Materials

Antibodies against phospho-S6 Ribosomal Protein (4857), phospho-mTOR (Ser2448) (2971), phospho-p42/44 Erk1/2 (Thr202/Tyr204) (9101), phospho-Akt (Ser473) (9271), mTOR (2972 and 4517), human platelet-derived growth factor BB (PDGF-BB) (8912) and RIPA cell lysis buffer (9806) were purchased from Cell Signaling Technology (Beverly, MA, USA). The Anti-cyclin D1 antibody [SP4] (16663) was from Abcam (Cambridge, MA, USA). β-actin (58522) antibody, diacylglycerol kinase inhibitor I (R59022), phosphatidic acid, and rapamycin were purchased from Cayman Chemicals (Ann Arbor, MI, USA). Secondary antibodies IRDye 680RD or 800CW were from LI-COR (Lincoln, NE, USA). Insulin-transferrin-selenium (41400045) and Dynabead™ Protein A beads (10001D) were from Thermo Fisher Scientific (Waltham, MA, USA). Protease and phosphatase inhibitors were from Bimake (Houston, TX, USA). All polyacrylamide gel casting, running, and transfer reagents and equipment were from Bio-Rad Laboratories (Hercules, CA, USA) or previously identified sources [35,50]. CyQuant cell proliferation assay kit and MTT Cell Viability Assay kits were from Life Technologies (Grand Island, NY, USA). Quantitative PCR arrays and SYBR green reagents were purchased from Real-Time Primers (Elkins Park, PA, USA) and Applied Biosystems (Grand Island, NY, USA), respectively.

### 4.2. Cell Culture

Human ASM cells were isolated from deidentified lung donors and cultured using complete F-12 media supplemented with 10% FBS, penicillin/streptomycin, HEPES buffer, CaCl_2_, L-Glutamine (Gibco), and NaOH between 2–6 passages, as described previously [51]. For western blotting, cells were grown on 12-well plates, and for Luciferase assay and CyQuant assay, cells were seeded onto 24-well or 96-well plates, respectively. Cells were cultured until fully confluent, then growth was arrested with serum-free F-12 medium containing 1% insulin-transferrin for 24, 48, or 72 h.

### 4.3. Viral Transduction of Human ASM Cells

Human ASM cells were transduced with lentiviral particles encoding luciferase reporter under the control of different promoters, including Serum Response Element (SRE), Nuclear Factor of Activated T-cells (NFAT), Signal Transducers for receptors of the Transforming Growth Factor beta (SMAD), and Activator Protein-1 (AP-1) (Qiagen, Germantown, MA, USA). Cignal Lenti Reporter Assays utilize a unique combination of multiple repeats of a transcription factor binding site and basic promoter elements to drive the expression of a reporter gene (firefly luciferase) coupled with lentiviral delivery. Human ASM cell cultures were infected with lentivirus as per the manufacturer’s recommendation, and as described previously [16,35]. Cells were used for experiments 24 h after lentiviral infection, as described below.

### 4.4. Luciferase (Luc) Reporter Assay

For luciferase assays, human ASM cells infected with different luciferase constructs were plated in 96-well plates at a seeding density of 10,000 cells per well. Cells were serum-starved for 24 h and then treated with DGK inhibitor I (30 µM) for 6 h. For experiments analyzing the activity of SRE-Luciferase, human ASM cells were pretreated with vehicle or rapamycin (10 nM) for 10 min, followed by stimulation with PA (0.5 μM–10 μM) for 12 h. Cells were lysed and luminescence was recorded following manufacturers’ protocol (Promega) using a FlexStation III plate reader, as described previously [16,52].

### 4.5. CyQuantCell Proliferation and MTT Assay

Human ASM cells were plated in a 96-well plate for CyQuant proliferation assay at a density of 5000 cells per well and maintained in complete Ham’s F-12 medium. Cells were serum starved for 24 h, and stimulated with growth factor PDGF (10 ng/mL), increasing concentrations of PA (0.1 µM–10 µM) or DGK inhibitor I (10 µM–30 µM) for 24, 48, or 72 h. In a select set of experiments, cells were treated with PA (0.1 µM–10 µM) 15 min before treating the cells with DGK inhibitor I (15 µM). After 24, 48, or 72 h treatment, cells were loaded with assay buffer containing CyQuant dye, and fluorescence intensity was measured as described previously using a FlexStation III plate reader [35]. For MTT cell viability assay, human ASM cells were plated in a 96-well plate at a density of 20,000 cells per well and maintained in complete Ham’s F-12 medium. Cells were serum starved for 24 h, and treated with vehicle, Triton X (1% vol.), or DGK inhibitor I (10 µM–30 µM) for 24 h. Cells were loaded with MTT reagent and solubilization of formazan was read by measuring absorbance using a FlexStation III plate reader.

### 4.6. Western Blotting

Human ASM cells were lysed in RIPA buffer Cell Signaling Technology) supplemented with protease inhibitor, phosphatase A inhibitor, and phosphatase B inhibitor (Bimake) at 4 °C for 30 min. Lysates were then mixed with Laemmli buffer (Bio-Rad) containing 10% β-mercaptoethanol. Lysates were boiled at 95 °C for 5 min, separated on SDS-PAGE, and transferred onto a nitrocellulose membrane. Target proteins were detected via incubation with antigen-specific primary antibodies overnight in TBST with 3% BSA. A secondary antibody conjugated with an infrared dye, either at 600 nm or 800 nm wavelength, was used to detect target protein using Odyssey infrared scanner (LI-COR Biosciences, Lincoln, NE, USA). Protein band intensity was quantified using Odyssey software, as described previously [51].

### 4.7. Co-Immunoprecipitation

Human ASM cells were cultured in 60 mm dishes and grown to confluence, then serum-starved for 24 h. Cells were pretreated with DMSO or rapamycin (10 nM) for 10 min then stimulated with vehicle, PDGF (10 ng/mL), or PA (5 or 10 µM) for 20 min at 37 °C. Cells were washed with ice-cold PBS for 2 min followed by harvesting of cell lysate using CHAPS (0.3%) lysis buffer, containing Tris HCl (25 mM, pH 7.4), NaCl (120 mM), EDTA (1 mM), and glycerol (5%), with final pH 7.4, on ice for 30 min. Cells were scraped off and transferred to new eppendorf tubes and spun down for 5 min at 2500 RPM at 4 °C. Protein concentration was determined using Pierce™ BCA Protein Assay Kit (Thermo Fisher Scientific, Waltham, MA, USA) and 100 μg of lysate was transferred to a new eppendorf tube and incubated with total mTOR antibody (1:50, vol:vol) overnight at 4 °C on a rotator. Antigen–antibody complex was incubated with Dynabead protein A beads (1.5 mg) for 1 h at 4 °C on a rotator and a magnet was used to separate the complex. Immunoprecipitate was washed three times with PBS containing 0.1% tween for 5 min on a rotator at 4 °C and resuspended in laemmli buffer. Immunoprecipitates were used for Bis-Tris polyacrylamide gel separation, then transferred onto a nitrocellulose membrane and immunoblotted as described above.

### 4.8. RNA Isolation and Real-Time PCR Array

Human ASM cells cultured on 6-well plates were treated with vehicle, PDGF (10 ng/mL), or PA (10 μM), with or without pretreatment with DGK inhibitor I for 72 h, and total RNA was harvested using Trizol as previously described [51,53,54]. Total RNA (1 µg) was reverse-transcribed to cDNA and gene expression was determined using a Real-Time PCR array for cell cycle genes (PAHS-020Z). cDNA was mixed with SYBR green master mix and added to 96-well plates containing primers for cell cycle genes and PCR was performed using Applied Biosystems real-time PCR machine. Raw Ct values were obtained using software-recommended threshold fluorescence intensity. RNA expression data were calculated as described previously using internal control gene GAPDH [51,53].

### 4.9. Murine Model of Allergen-Induced Asthma

In this study, we used C57BL/6 (wild type) and DGKζ knockout (KO) mice. C57BL/6 mice were purchased from Charles River Laboratories and DGKζ KO mice were generated as described previously [31,55,56]. Wild type and KO mice (8–10 weeks) were subjected to an HDM allergen challenge, as described previously [57,58]. Briefly, mice were challenged with HDM (25 μg in 35 μL PBS/mouse/dose) via intranasal instillation for 5 days a week for a total of 3 weeks. Twenty-four hours post-final dosing and HDM challenge, mice were euthanized and processed for excision of lung tissues for histopathological analyses. All animal procedures were reviewed and approved by the Institutional Animal Care and Use Committee (IACUC) at University of Pennsylvania.

Histological evaluation of lung tissues: Murine lungs fixed with formalin buffered saline were embedded in paraffin, and 5 μm thick sections were obtained. Tissue sections were mounted on SuperfrostTM Plus microscopic slides (Fisher Scientific, Waltham, MA, USA). Then, tissue sections were deparaffinized with xylene and rehydrated in Shandon Varistain Gemini ES Autostainer. Antigen retrieval was performed using pH 6.0 10 mM Citrate Buffer at 98 °C for 20 min with DAKO PT Link. Post-retrieval slides were stained with hematoxylin and eosin (H&E) (Biocare Medical, Pacheco, CA, USA). A set of sections were stained with Periodic acid-Schiff (PAS) (Poly Scientific R&D Corp, Bayshore, NY, USA) as per the manufacturer’s protocol for standard histopathological evaluations. Some sections were further processed for immunohistochemical staining using anti-α-smooth muscle actin (diluted 1:1000, Abcam, Cambridge, MA, USA), and anti-Ki-67 (diluted 1:100, Abcam, Cambridge, MA, USA) primary antibodies for 30 min. Biotinylated anti-rabbit or anti-rat secondary antibodies (diluted 1:200, Vector Laboratories, Burlingame, CA, USA) and ABC-HRP complexes (Vector Laboratories, Burlingame, CA, USA) were applied following primary antibodies and incubated at room temperature for 30 min. Antibody binding was visualized using ImmPACT DAB EqV Peroxidase (HRP) Substrate (Vector Laboratories, Burlingame, CA, USA) and then counterstained with hematoxylin counterstain in Varistain Gemini ES Autostainer. Tissue sections were covered with coverslips mounted with Permount Mounting Medium. All images were acquired using a brightfield light microscope. Image analysis was performed using NIH Image J 1.53t software (Java 1.8.0_112 (64-bit)).

### 4.10. Statistical Analysis

Data are presented as mean ± SEM values from *n* experiments, where *n* represents distinct donors or number of animals. PAS staining was quantified by counting the number of PAS+ cells in each airway using a brightfield microscope. The number of PAS+ cells was normalized to the total area of the airway. For all other histology data, the staining particle count or intensity was obtained by an image deconvolution method using Image J. Staining intensity was normalized to the total area of the airway. Individual data points from a single experiment were calculated as the mean value from three replicate observations for CyQuant assay and luciferase assay. Data from ASM growth assay and luciferase assay were calculated and reported as fold change from the basal or vehicle-treated group. For immunoblot analyses, band intensities were normalized to values determined for β-actin and compared among stimuli and experimental groups. Statistically significant differences among groups were assessed by either one-way or two-way ANOVA with Bonferroni post-hoc analysis or Student’s t-test using GraphPad Prism 6 software (Graphpad, La Jolla, CA, USA), with values of *p* < 0.05 sufficient to reject the null hypothesis.

## Figures and Tables

**Figure 1 ijms-23-11868-f001:**
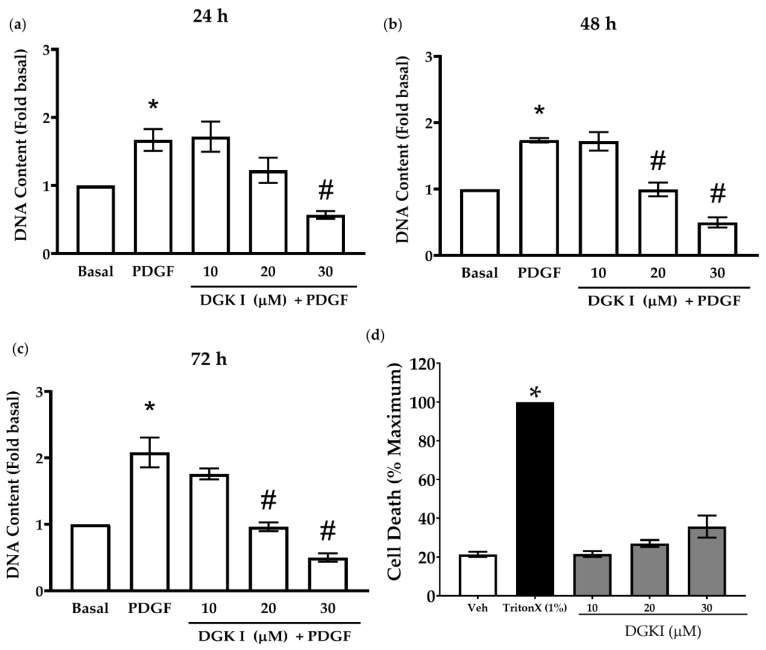
Effect of DGK inhibition on PDGF-induced ASM cell proliferation. Human ASM cells were treated with different concentrations of DGKI (10 µM–30 µM) for 15 min, followed by PDGF (10 ng/mL) stimulation for (**a**) 24 h, (**b**) 48 h, and (**c**) 72 h. Cell proliferation was assessed by measuring total DNA content using CyQuant assay and normalized to basal condition. Human ASM cells were treated with DGKI (10 µM–30 µM) for 24 h and cell viability was assessed by MTT assay (**d**). Data were normalized to maximum cell death by positive control TritonX (1%). Data above are mean ± SEM from *n* = 5–6 independent donors (**a**–**c**) and *n* = 3 independent donors (**d**). * *p* < 0.05 compared to basal, # *p* < 0.05 compared to PDGF using one-way ANOVA with Bonferroni post-hoc analysis.

**Figure 2 ijms-23-11868-f002:**
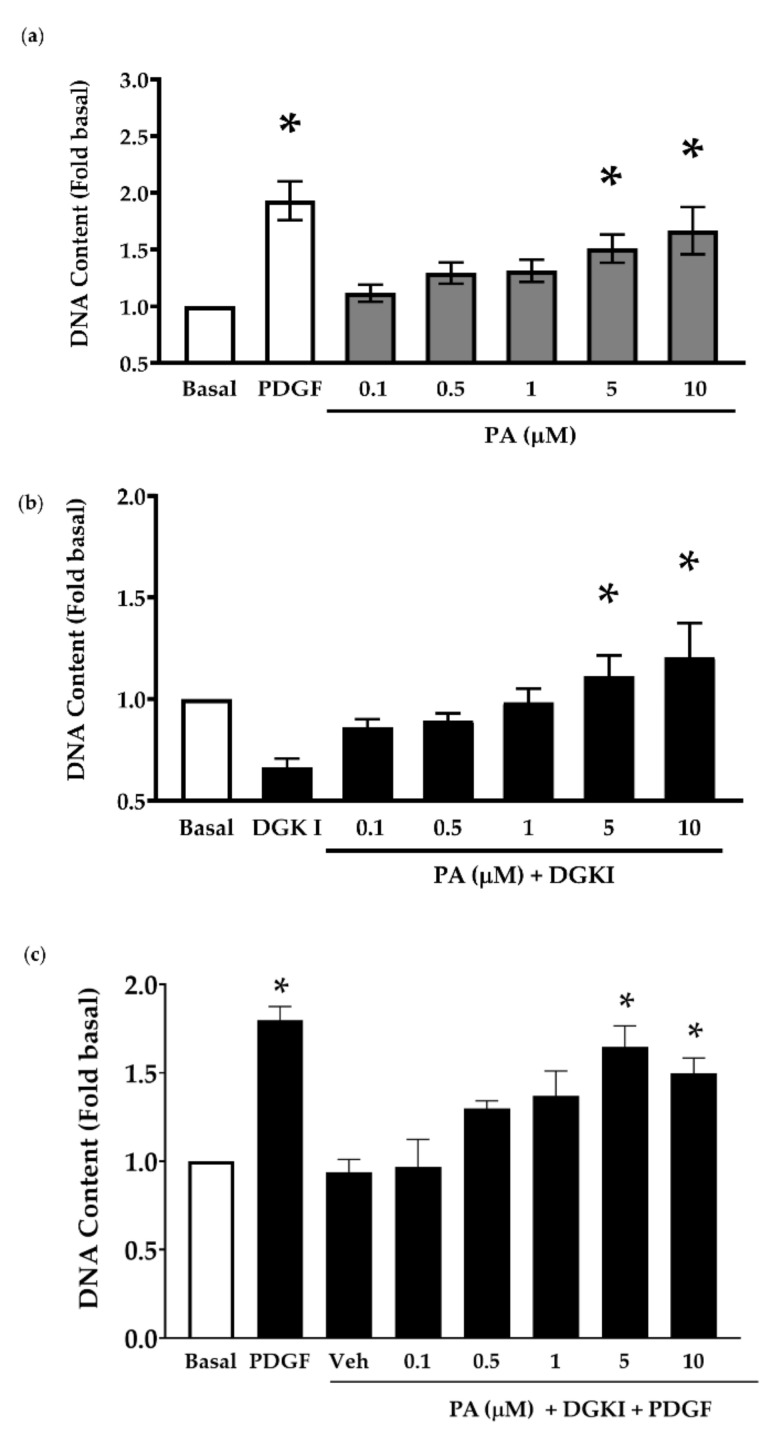
Effect of phosphatidic acid (PA) on ASM cell proliferation. (**a**) Human ASM cells were stimulated with PA (0.1 μM–10 μM) alone, (**b**) in the presence of DGKI (15 μM), or (**c**) in the presence of DGKI (15 μM) plus PDGF (10 ng/mL) for 72 h. Cell proliferation was evaluated using CyQuant assay and data normalized to basal condition. (**a**,**c**) Treatment with PDGF (10 ng/mL) was used as a positive control. Data above are mean ± SEM from *n* = 3–6 independent donors. * *p* < 0.05 compared to (**a**,**c**) basal or (**b**) DGKI using one-way ANOVA with Bonferroni post-hoc analysis. PA: Phosphatidic acid.

**Figure 3 ijms-23-11868-f003:**
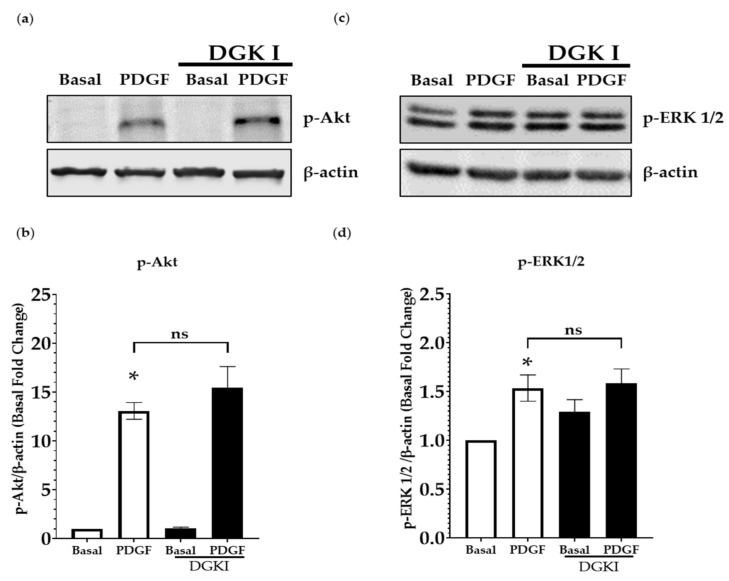
Effect of DGK inhibition on phosphorylation of Akt and ERK1/2. Human ASM cells were pretreated with DGKI (30 µM) for 15 min followed by stimulation with PDGF (10 ng/mL) for 15 min. (**a**,**b**) Cells were lysed and immunoblotted for p-Akt and (**c**,**d**) p-ERK1/2. p-Akt and p-ERK1/2 densitometry are represented as fold change to basal conditions. Data above are mean ± SEM from *n* = 5–7 independent donors. * *p* < 0.05 Basal vs. PDGF using one-way ANOVA with Bonferroni post-hoc analysis.

**Figure 4 ijms-23-11868-f004:**
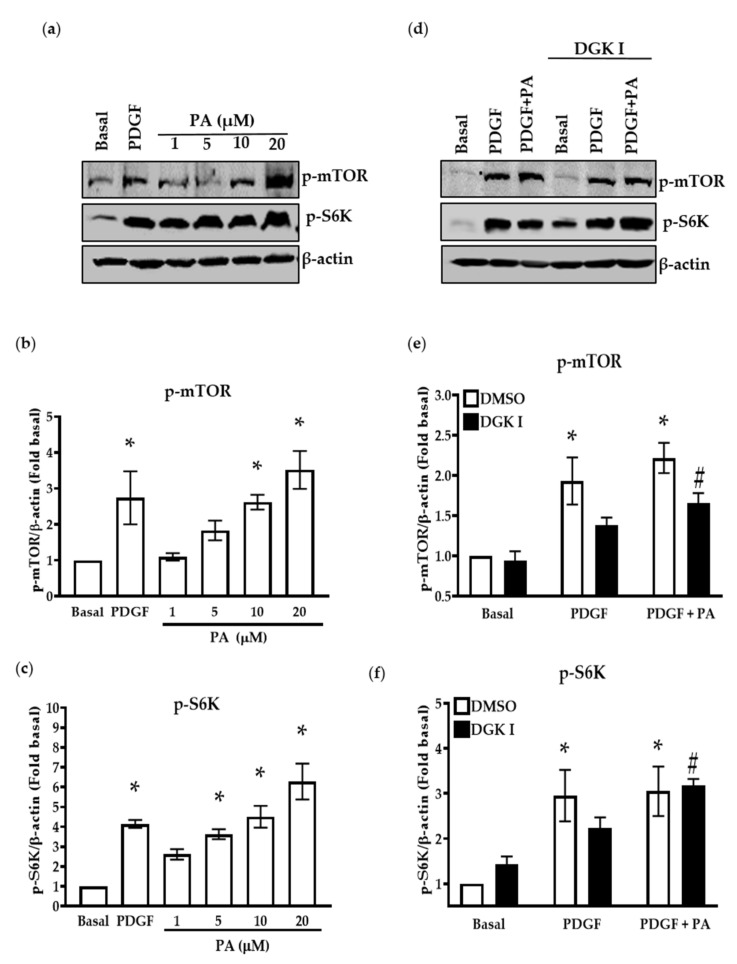
Effect of DGK inhibition and exogenous PA treatment on phosphorylation of mTOR and S6K. Human ASM cells were stimulated with PDGF (10 ng/mL) or PA (1 µM–20 µM) for 10 min. (**a**) Cells were lysed and immunoblotted for p-mTOR, p-S6K, and β-actin. Graphical representation of (**b**) p-mTOR and (**c**) p-S6K densitometry data from blots normalized to basal condition. (**d**) Human ASM cells were pre-treated with DGKI (30 µM) for 15 min and then stimulated with PDGF (10 ng/mL) or PDGF (10 ng/mL) with PA (10 µM) for 10 min, lysed, and immunoblotted for p-mTOR, p-S6K, and β-actin. Graphical representation of (**e**) p-mTOR and (**f**) p-S6K densitometry data from blots normalized to basal condition. Data above are mean ± SEM from *n* = 4–6 independent donors. * *p* < 0.05 compared to basal, # *p* < 0.05 DGKI (PDGF vs. PDGF + PA) using one-way ANOVA with Bonferroni post-hoc analysis. PA: Phosphatidic acid.

**Figure 5 ijms-23-11868-f005:**
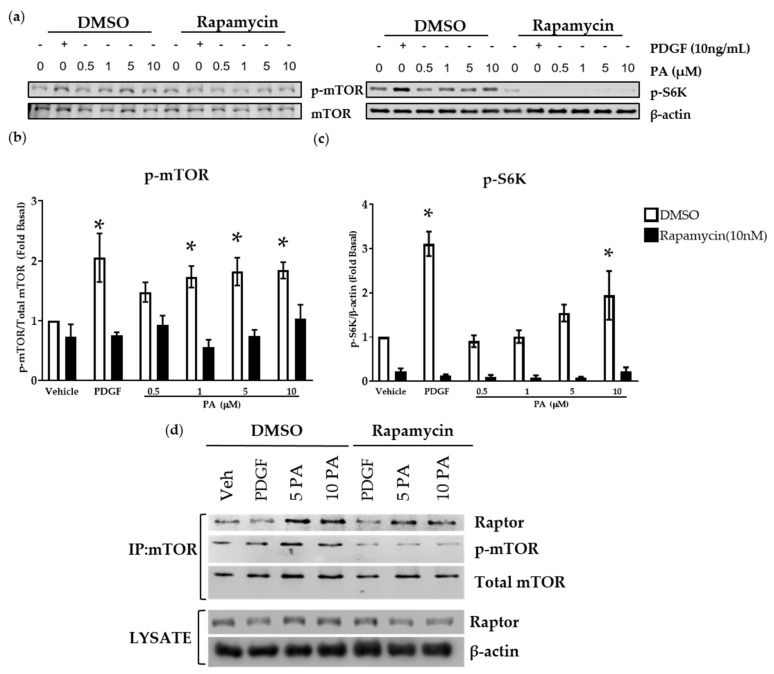
Effect of PA on the formation of mTOR complex 1. Human ASM cells were treated with DMSO or rapamycin (10 nM) for 10 min followed by stimulation with vehicle, PDGF (10 ng/mL), or PA (0.5 μM-10 μM). Cells were lysed and immunoblotted for (**a**,**b**) p-mTOR; (**a**,**b**) total mTOR; (**a**,**c**) p-S6K; and (**a**,**c**) β-actin. Graphical representation of mean ± SEM of the (**b**) p-mTOR and (**c**) p-S6K densitometry data from blots normalized to basal condition. hTERT ASM cells were treated with DMSO or rapamycin (10 nM) for 10 min followed by stimulation with PDGF (10 ng/mL), or PA (5 µM or 10 µM) for 20 min. Cells were lysed and immunoprecipitated using a total mTOR antibody. (**d**) Lysates and immunoprecipitates were immunoblotted for raptor, p-mTOR, mTOR, and β-actin. (**a**–**c**) Data above are representative from *n* = 4–7 independent donors and (**d**) *n* = 3 independent experiments. * *p* < 0.05 compared to vehicle using one-way ANOVA with Bonferroni post-hoc analysis.

**Figure 6 ijms-23-11868-f006:**
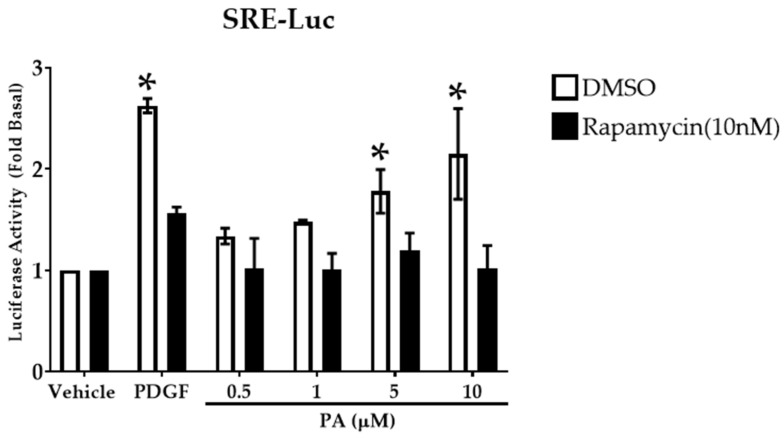
Effect of PA on transcription factor activation. Human ASM cells were transduced with lentivirus particles expressing luciferase under the control of multiple transcription factors and pre-treated with DMSO or rapamycin (10 nM) for 10 min followed by stimulation with PDGF (10 ng/mL) or PA (0.5 μM–10 μM) for 12 h. Luciferase activity was determined by measuring luminescence. Data are represented as fold-change to basal condition. Data above are mean ± SEM from *n* = 3–6 independent donors. * *p* < 0.05 compared to vehicle using one-way ANOVA with Bonferroni post-hoc analysis. PA treatment promoted SRE-Luc activity in human ASM cells among all the transcription factors tested. PA-induced SRE activation is mediated via mTOR signaling.

**Figure 7 ijms-23-11868-f007:**
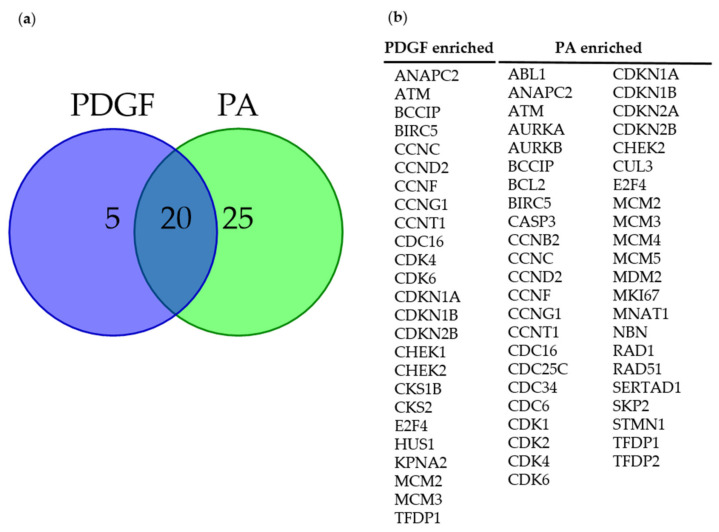
Effect of DGK inhibition on cell cycle genes. Human ASM cells were treated with PDGF (10 ng/mL) or PA (10 µM) for 72 h, lysed, and RNA was harvested and used for cell cycle gene analysis by qRT-PCR. (**a**) A pie chart representing the number of genes that are upregulated more then two folds compared to vehicle-treated cells; (**b**) list of genes differentially enriched in PDGF and PA treated cells. Human ASM cells were pretreated with DGKI (30 µM), followed by stimulation with PDGF (10 ng/mL) or PA (10 µM) for 72 h.

**Figure 8 ijms-23-11868-f008:**
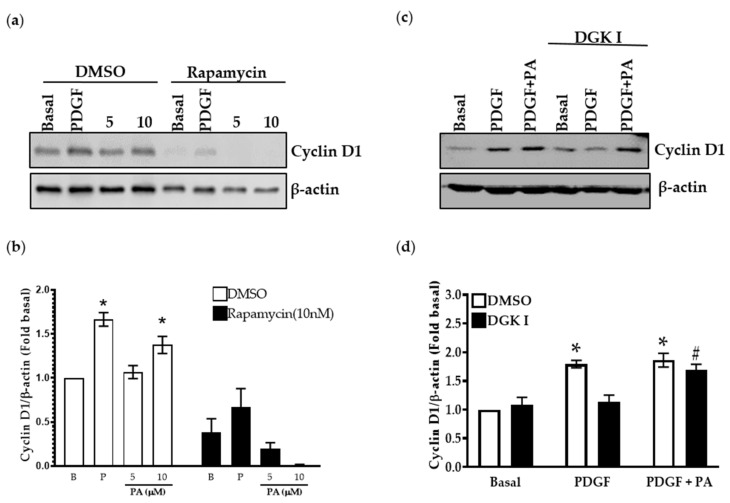
Effect of DGK and rapamycin inhibition on cyclin D1 expression. (**a**,**b**) Human ASM cells were pretreated with rapamycin (10 nM) for 10 min, followed by stimulation of PA (5 µM or 10 µM) for 24 h, lysed, and immunoblotted for cyclin D1; (**c**,**d**) Human ASM cells were pretreated with DGKI (30 µM) followed by stimulation with PDGF (10 ng/mL), or PDGF (10 ng/mL) with PA (10 µM) for 24 h, lysed and immunoblotted for cyclin D1. Graphical representations of densitometry data from blots are normalized to basal conditions. Data above are mean ± SEM from *n* = 4–7 independent donors. * *p* < 0.05 compared to basal, # *p* < 0.05 DGKI (PDGF vs. PDGF + PA) using one-way ANOVA with Bonferroni post-hoc analysis. PA: Phosphatidic acid.

**Figure 9 ijms-23-11868-f009:**
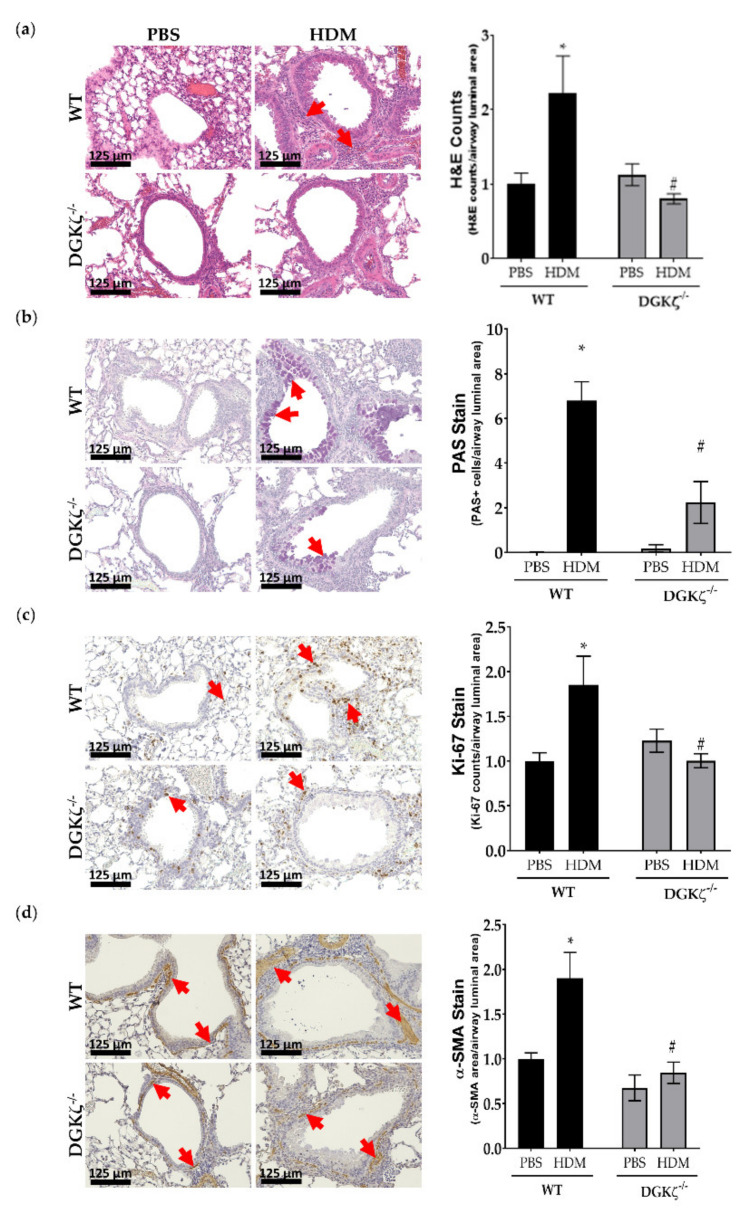
Effect of lack of DGKζ on development of allergen-induced airway remodeling. DGKζ^−/−^ mice were challenged with HDM for 3 weeks and lung tissue was collected. Tissue was formalin-fixed and paraffin-embedded. Tissue sections were stained with (**a**) H&E, (**b**) PAS, (**c**) Ki-67, and (**d**) α-SMA. H&E cellular counts, PAS+ cells, Ki-67+ cellular counts, and α-SMA area were obtained and normalized to airway luminal area. These values were further normalized to the average value of wild type (WT) PBS group. Data above are mean ± SEM from *n* = 3–4 mice per treatment group. * *p* < 0.05 compared to WT PBS, # *p* < 0.05 compared to WT HDM using two-way ANOVA with Bonferroni post-hoc analysis.

**Figure 10 ijms-23-11868-f010:**
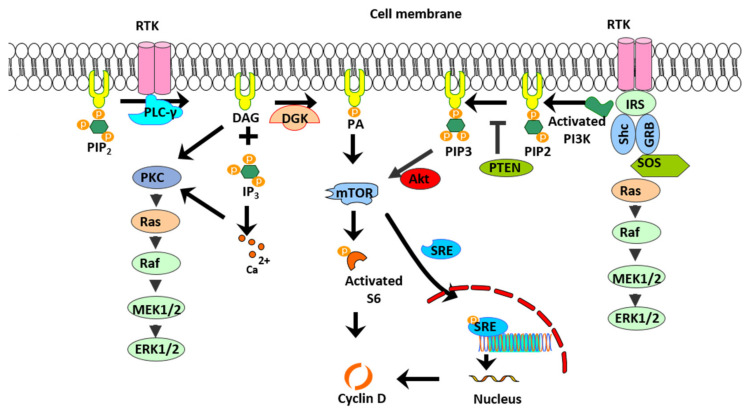
Model depicting PA-mediated ASM cell proliferation. PA generated by DGK leads to increased formation and activity of mTORC1. mTOR increases activity of S6K and transcription factor SRE, resulting in increased transcription and translation of cell cycle progression genes, including cyclin D1. DGK inhibition attenuates mTOR signaling by depleting levels of PA in ASM cells.

**Table 1 ijms-23-11868-t001:** List of genes upregulated by PDGF (10 ng/mL), downregulated by DGKI (30 µM), and rescued by PA (10 µM) in human ASM cells stimulated for 72 h.

Phase of Cell Cycle	Protein	Up Regulated by PDGF	Down Regulated by DGK	Rescued by PA
**G1 Phase & G1/S Transition**	Anaphase promoting complex subunit 2	*ANAPC2*		
	Cyclin-dependent kinase 6	*CDK6*	*CDK6*	*CDK6*
**S Phase & DNA Replication**	Minichromosome maintenance complex component 2	*MCM2*		
	Minichromosome maintenance complex component 3	*MCM3*	*MCM3*	
**G2 Phase & G2/M Transition**	BRCA2 and CDKN1A interacting protein	*BCCIP*	*BCCIP*	
	Baculoviral IAP repeat containing 5	*BIRC5*	*BIRC5*	
	Cyclin G1	*CCNG1*		
	Cyclin T1	*CCNT1*	*CCNT1*	*CCNT1*
	CDC28 protein kinase regulatory subunit 1B	*CKS1B*		
	CDC28 protein kinase regulatory subunit 2	*CKS2*	*CKS2*	*CKS2*
	Karyopherin alpha 2	*KPNA2*	*KPNA2*	*KPNA2*
**M Phase**	Cyclin F	*CCNF*	*CCNF*	*CCNF*
	Cell division cycle 16 homolog (S. cerevisiae)	*CDC16*		
**Cell Cycle Checkpoint & Cell Cycle Arrest**	Cyclin-dependent kinase inhibitor 1A (p21, Cip1)	*CDKN1A*	*CDKN1A*	*CDKN1A*
	Cyclin-dependent kinase inhibitor 1B (p27, Kip1)	*CDKN1B*	*CDKN1B*	*CDKN1B*
	Cyclin-dependent kinase inhibitor 2B (p15, inhibits CDK4)	*CDKN2B*	*CDKN2B*	*CDKN2B*
	CHK1 checkpoint homolog (S. pombe)	*CHEK1*		
	CHK2 checkpoint homolog (S. pombe)	*CHEK2*		
	HUS1 checkpoint homolog (S. pombe)	*HUS1*	*HUS1*	
**Regulation of the Cell Cycle**	Cyclin C	*CCNC*	*CCNC*	*CCNC*
	Cyclin D2	*CCND2*	*CCND2*	*CCND2*
	Cyclin-dependent kinase 4	*CDK4*	*CDK4*	
	E2F transcription factor 4, p107/p130-binding	*E2F4*		
	Transcription factor Dp-1	*TFDP1*	*TFDP1*	*TFDP1*
**Negative Regulation of the Cell Cycle**	Ataxia telangiectasia mutated	*ATM*		

## Data Availability

Not applicable.

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
