# Peer review of "Regulation of Airway Smooth Muscle Cell Proliferation by Diacylglycerol Kinase: Relevance to Airway Remodeling in Asthma"

_ijms, 2022, doi:10.3390/ijms231911868_

Round 1
Reviewer 1 Report
This study explores the less understood role of Diacylglycerol kinase (DGK) and Phosphatidic acid (PA) in the regulation of ASM cell proliferation. Although the role of PA has been linked to various smooth muscle cell physiological functions like actin polymerization, cell migration and mitogenesis, this study demonstrates that PA is a pro-mitogenic second messenger in ASM cells. The authors suggest that PA can be a novel regulator of ASM proliferation, and DGK inhibition can be a promising therapeutic approach to mitigate ASM remodeling in asthma.
Minor comments:
i. 1. Please include the role of PA in other cell types in the introduction section.
ii. In Figure 1, please include:
a. the statistical significance between Basal and PDGF treated groups seem mssing.
b. DGKI + PDGF on graphs for DGKI pre-treated groups followed by PDGF stimulation for better understanding for readers.
iii. 2. Was there any difference in DNA content observed between 24h, 48h and 72h in DGKI groups in Figure 1?
iv. 3. Labeling of lanes in western blots (B, P) is confusing in Figures 3, 4, 8. Please label as basal, PDGF instead.
v. 4. Please include the scale bar for tissue sections in Figure 9.
Author Response
Responses are in the attached document

Reviewer 2 Report
Comments to the Authors (IJMS-1897708)
The authors demonstrated that diacylglycerol kinase (DGK)contributes to a phenotypic switching of airway smooth muscle (ASM) cell froma contractile to proliferative states and, subsequently, promotes asthma. These data are of interest, however, there are some concerns in the present form. Overall, it is difficult to be enthusiastic about publication in the present form.
1) Figure 1: The authors demonstrated the effect of DGKI on PDGF-induced ASM cell proliferation by measuring DNA content. It should be also showed the viability of ASM cells using such as MTT assay.
2) Figure 2: Although the authors claimed that exogenous PA mitigated the effect of DGKI on ASM cell proliferation, it is better to show whether exogenous PA reduces DGKI-dependent inhibition of ASM cell proliferation under PDGF stimulation. Also, other experiments.
3) Figure 4 and 5: the amounts of p-S6K in PA (1~10µM) were quite different between Figure 4 and 5. In Figure 4, the amounts of p-S6K in 5~10µM PA were increased by similar level as PDGF, but those in Figure 5 were about half. In contract, the amounts of p-mTOR in PA (Figure 4) were concentration-dependently increased, but not in Figure 5. In addition, immunoblot signal of p-mTOR from immunoprecipitated from DMSO was not increased in PDGF condition (used as a positive control). These results indicated that there were some side effects under these experimental conditions.
4) Figure 6: As described above, the authors should address the issue under DMSO conditions.
5) Figure 9: In Materials and Methods, it was mentioned that DGKζ knockout (KO) mice was used in this study. Is this KO mouse a smooth muscle specific KO such as Myh11-Cre DGKζfl/fl? The authors had previously reported that DGKζ plays a central role in the asthma through affecting immune cells (ref 20). In this previous report, authors generated several DGKζ KO mice including Myh11-Cre DGKζfl/fl.
6) Figure 9d: Another issue is thatimage data were poor and not clear. In Figure 9d, it is critical to show ASM cell proliferation in HDH challenged lung. Immunofluorescence staining might be better resolution in order to detect alpha-SMA (ASM cells).
Author Response
Responses to the comments are in the attached document.

Round 2
Reviewer 2 Report
Comments to the Authors (IJMS-1897708)
Overall the manuscript is much improved from the previous submission. The authors have addressed my concerns with new experiment and textual changes, as required. I am now entirely convinced that this article should be published in theIJMS.